# I Am Better Than Others: Waste Management Policies and Self-Enhancement Bias

Yihan Zhao [1,2,†], Rong Chen [1,2,†], Mitsuyasu Yabe [3], Buxin Han [1,2] and Pingping Liu [1,2,*]

1   CAS Key Laboratory of Mental Health, Institute of Psychology, Chinese Academy of Sciences, 16 Lincui Road, Beijing 100101, China; seventeenzyh@163.com (Y.Z.); 20132501074@m.scnu.edu.cn (R.C.); hanbx@psych.ac.cn (B.H.)
2   Department of Psychology, University of Chinese Academy of Sciences, 19A Yuquan Road, Beijing 100049, China
3   Laboratory of Environmental Economics, Department of Agricultural and Resource Economics, Faculty of Agriculture, Kyushu University, 744 Motooka Nishi-ku, Fukuoka 819-0395, Japan; yabe@agr.kyushu-u.ac.jp
*   Correspondence: liupp@psych.ac.cn; Tel.: +86-10-64879731
†   These authors contributed equally to this work.

**Abstract:** Waste source separation has been a social dilemma globally with a low participation rate. This research attempted to solve this dilemma by exploring the effect of mandatory (versus voluntary) policies on waste separation from the perspective of the self-versus based on deterrence theory and self-enhancement motivation. Hypothetical scenarios were used to demonstrate the effectiveness of mandatory policies and self-enhancement bias for residents ($n$ = 589) and adolescents ($n$ = 121). Study 2 was performed to replicate the findings of Study 1 with a no-implementation policy condition, and Study 3 extended the findings to adolescents. We found robust self-enhancement bias, where participants perceived themselves to be better than others in both willingness to perform and attitudes toward waste separation behavior. Specifically, participants tended to perceive themselves to perform waste separation well when policy compliance was voluntary, but they tended to perceive others to perform well when policy compliance was mandatory with supervision. These findings highlight the impact of mandatory policy with supervision and self-enhancement bias in waste management. The present studies provide substantial evidence and implications for the necessity of supervision in mandatory policy implementation.

**Keywords:** self-enhancement bias; mandatory policy; supervision; waste separation

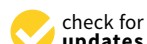



## 1. Introduction

Waste separation has emerged as a major challenge to environmental management worldwide. There is less available land for large-scale waste disposal in landfills and incinerators. According to the World Bank [1], the amount of global waste will increase from 2 billion tons in 2016 to 3.4 billion tons in 2050 with rapid population growth and urbanization. The growth of waste levels has become a critical global issue because it poses a threat to human health and causes environmental pollution. Separating waste at the source has become a vital element of waste management strategies for reducing waste in many countries [2–4]. Given the urgency of waste disposal problems, many municipalities have implemented waste separation policies to reduce landfill or incineration use [5–7].

The types of environmental regulation or regional policies (e.g., mandatory vs. voluntary) may lead to the illustration of different waste separation or recycling rates [8–14]. For instance, the recycling rates range from 1% in Louisiana (USA) to 56.1% in North Kevesten (UK) where the EU's Landfill and Waste Framework directives were implemented [7,15]. The recycling rate in Ireland increased from 11% in 2001 to 33% in 2011 since the introduction and implementation of the Landfill Directive 1999/31/EC, the WEEE directive 2002/96/EC, and the Waste Framework Directive 2008/98/EC [5]. The recycling rate

reached 85% within three months when New Jersey, U.S. mandated recycling and promoted a curbside pick-up program [2]. In the early 1990s, China started to implement waste separation policies to promote the participation rate of waste separation. A new national-level policy was issued in 2017, and 46 pilot cities were selected to implement the new mandatory waste separation policy "Implementation plan on the household solid waste classification system." Shanghai became the leading city in 2019 to enforce the mandatory waste separation policy [16–19]. It concluded that the mandatory policy could reduce household food waste. This showed that policies would affect the effectiveness of waste management.

Although some waste management strategies have been implemented, waste source separation has not been successful due to a lack of public participation thus far [3,8,20]. Despite numerous public awareness campaigns conducted over the years, public participation in waste separation is still at a low level [4,17,21–24]. Most residents express a willingness to separate waste, but only a few perform the task on a daily basis [20,22,25,26]. Thus, it is a key question to promote public participation in waste separation.

Moreover, waste source separation efficiency depends on the concerted efforts of social members [8,17,21,27]. Similar to other prosocial behaviors, such collective action is vulnerable to the free-rider problem [28,29], which could be solved through externally enforced regulations such as mandatory policies [30]. Others' behaviors greatly affect our actions because we are strongly influenced by others [31,32]. Solving the commons dilemma requires cooperation with oneself and others. People are conditional cooperators, and they contribute only if others also do so [33,34]. However, it is unclear how people perceive themselves and others in terms of policies and behaviors of waste source separation.

Thus, the aim of the current research is to investigate the influence of mandatory policies regarding the discrepancies in residents' perception of their own and others' waste separation behaviors. In particular, we first review prior studies on mandatory policies, waste separation, and self–others discrepancies. Then, we introduce our hypotheses. Third, we present three experimental studies on participants' perception of their own and others' waste separation. Finally, our research findings and policy implications are discussed.

## 2. Literature Review

### 2.1. Waste Separation Policies

An effective environmental regulation system is a crucial driver to the improvement of waste source separation. Based on the literature [10,35,36] and the situation in China, we conceive that waste separation policies can be classified into three types: mandatory policy, voluntary policy, and no-implementation policy. Mandatory policies refer to the rules governing behavior, and policymakers design a package of commonly agreed standards, penalties, supervision, and enforcement when mandates are used [35,36]. Previous studies suggest that mandatory policies play key roles in promoting waste separation [20,37–39]. For instance, in July 2019, Shanghai took the lead in formulating and implementing a mandatory waste separation policy via supervisory guidance, penalties, regulated disposal times, and others [40], and its waste separation rate increased from 15% to 80% [41]. Many educators and funding agencies share the belief that mandatory policies increase public participation and compliance with waste separation regulation.

The mechanism of efficient mandatory policies can be explained by deterrence theory, which proposes that mandatory policies can act as a deterrent [20,42]. Deterrence theory assumes that individuals make rational decisions through a cost-benefit analysis. If the expected cost of offending behaviors outweighs the expected benefits, individuals are less likely to engage in offending behaviors. These offending behaviors could be deterred by enhancing three elements of punishment, including certainty, severity, and celerity [42]. Certainty refers to the possibility that illegal waste separation behavior will be discovered and punished, severity reflects the harshness of punishment incurred, and celerity refers to the timeliness of penalty execution [43,44].

Although theoretically efficient, some studies suggest that mandatory policies might not always be effective. For instance, following the example of Shanghai, the city of Zhengzhou implemented a waste separation policy in December 2019. However, the separation rate in Zhengzhou showed less improvement (from 10.1% to 17.8%) [24]. Some studies also found a weak relationship between policies and waste separation behavior [17] and showed that a more stringent waste policy might increase the illegal disposal of waste [45].

Mandatory policies cannot work in some situations. One possibility could be an absence of supervision or compulsory actions to guarantee the implementation of policies. These can be regarded as voluntary waste separation policies which have been implemented without intervention by the government. Waste separation is strongly recommended by the government. However, if somebody did not separate waste, he or she will not be punished. A non-compulsory or voluntary-based policy could not effectively restrain residents' free-riding behavior, so it had little effect on their waste separation [20]. In contrast, several studies have found that volunteer supervision significantly facilitates waste separation [46–48]. This evidence suggested that voluntary policies had a weak impact on waste separation unless supervision or feedback was implemented. Therefore, the effectiveness of waste separation may depend on a country's enforcement, sanctions, and supervision.

In China, 46 pilot cities were selected to implement mandatory waste separation policies. The policy facilitated waste separation with strict supervision in some cities, such as Shanghai [41], while it influenced weakly in other cities, like Zhengzhou, without supervision [24]. There was even a no waste separation policy implemented in many cities such as Shangrao of Jiangxi Province. Taken together, different types of waste separation policies (i.e., a mandatory policy with supervision, voluntary policy without supervision, and no-implementation policy) showed different effects on actual waste separation behavior.

Here, we investigate the effectiveness of mandatory, voluntary, and no-implementation policies in encouraging waste separation behavior through three studies. Based on prior studies and intuitive reasoning, we expected that mandatory policies with supervision would increase the likelihood of waste separation behavior but voluntary without supervision could not. We also predicted that a no-implementation policy would have insignificant impacts on waste separation behavior.

### 2.2. Self-Others Discrepancies

Recyclables become contaminants when someone engages in incorrect waste separation [46]. Some people are reluctant to sort waste because they might feel that others engage in free riding, and thus that their contributions would be useless. This is a social dilemma. Tam and Chan [32] proposed that people who were concerned about environmental issues were reluctant to adjust their behavior due to a fear of being exploited by free riders. Our perception of others' behavior may affect our own level of cooperation. Solving the commons dilemma requires cooperation, and the behavior of others then becomes an important consideration [34]. However, it is unclear how people perceive themselves and others in terms of waste separation behavior and whether self–others discrepancies exist for different types of policies.

Self–others discrepancies refer to the difference between self-perception and how an individual perceives others [49]. According to the theory of social comparison [50], people possess a drive to engage in self-evaluation by comparing themselves to others. The tendency to compare ourselves to others is a fundamental, ubiquitous, and robust human proclivity [51].

Human beings hold an excessively positive view of themselves and of things associated with the self when they perceive themselves and others. People tend to believe that they are better than others in many favorable characteristics, such as positive traits and behaviors [52]. This tendency is termed the self-enhancement bias or better-than-average



effect (BTAE), which can be explained by motivational mechanisms [53]. People are motivated to possess a positive self-concept to maintain their self-esteem. This cognitive self-enhancement bias has been documented across many socially valued dimensions [53–55].

However, to our knowledge, there are a few studies on self-enhancement bias related to pro-environmental behaviors [56]. Bergquist [57] found that people perceive themselves as more pro-environmental than others in terms of conserving energy, recycling, and so on. Nevertheless, little direct evidence indicates whether self-enhancement bias exists for the domain of waste separation in the context of mandatory and voluntary policies. According to prior studies [56,57], we predicted that people assess themselves as more likely to sort waste than others in both *mandatory* and *voluntary conditions* and to hold a more positive attitude toward waste separation than others.

Taken together, the following hypotheses are proposed.

**Hypothesis 1a:** *A mandatory policy with supervision increases the likelihood of engaging in waste separation behavior.*

**Hypothesis 1b:** *Both voluntary policy and no-implementation policy conditions cannot facilitate engagement in waste separation behavior.*

**Hypothesis 2:** *People perceive themselves as more engaged in waste separation behavior than others.*

**Hypothesis 3:** *People perceive themselves to hold a more positive attitude toward waste separation than others.*

*2.3. The Present Studies*

Age is considered as an important factor that influences people's perception and implementation of policies on waste separation [58–61]. Although previous research has demonstrated that children take a strong stance toward protecting the natural environment, it was found that adolescents engage less in pro-environmental behaviors [58]. Adolescents are an important population to target with pro-environment behaviors [59]. Not only are adolescents high consumers of natural resources, but they are also the adult consumers of the future [60]. Therefore, policies targeting the pro-environmental behavior of adolescents were implemented on a national level in many countries [61]. Adolescents can acquire information via formal education or by participating in environmental education programs (e.g., waste separation), and they can radiate waste separation behavior in their families. Thus, we explored the effectiveness of mandatory policies and self-enhancement bias for both adults (residents) and adolescents (see Table 1).

**Table 1.** Research framework and designs.

| | Policy | | | Person | | Participants | Sample Size |
|---|---|---|---|---|---|---|---|
| | **Mandatory** | **Voluntary** | **No-Implementation** | **Self** | **Others** | | |
| Study 1 | √ | √ | × | √ | √ | Residents | 240 |
| Study 2 | √ | √ | √ | √ | √ | Residents | 349 |
| Study 3 | √ | √ | × | √ | √ | Adolescents | 121 |

We conducted three studies to test the important roles of mandatory policies in waste separation behavior from the perspectives of self–others discrepancies. We used hypothetical scenarios to demonstrate the effectiveness of mandatory policies and self-enhancement bias for residents and adolescents. We expected that the participants predicted being more willing to engage in waste separation behavior under a mandatory policy with supervision than under a voluntary policy. To replicate the findings of Study 1, Study 2 added a baseline condition (no-implementation policy), and Study 3 extended the results to adolescents.

## 3. Study 1

Study 1 aimed to examine how people perceive themselves and others to sort waste under mandatory and voluntary policies. We expected the existence of self-enhancement bias in the field of waste separation.

### 3.1. Materials and Methods

3.1.1. Research Design

This study involved a 2 (policy: *mandatory* vs. *voluntary*) × 2 (person: *self* vs. *others*) between-participants design (see Table 2). According to prior research [62,63], we designed the following hypothetical scenarios. In the *mandatory condition*, the participant imagined that waste separation is made mandatory and intervened in (e.g., supervision, feedback and punishment) by the government. In the *voluntary condition*, the scenario involved a waste separation policy without intervention or supervision. The participant imagined that waste separation is strongly recommended by the government but remains voluntary. For the two-person condition, the *self condition* assessed the participants' own attitudes toward and willingness to engage in waste separation behavior. The *others condition* assessed the participants' predictions of others' attitudes toward and willingness to engage in waste separation behavior. Participants were randomly assigned to the four conditions (*mandatory–self, mandatory–others, voluntary–self, and voluntary–others*), and each participant completed the task for one condition.

**Table 2.** Materials and measurement in Study 1.

| Conditions (*self/others* version for the hypothetical scenario): | |
| --- | --- |
| *Mandatory* | Imagine that the municipal government has published and implemented a waste separation policy. Both public education and supervision and punishment strategies are adopted. |
| *Voluntary* | Imagine that the municipal government has published and implemented a waste separation policy involving public education but no supervision or punishment strategies. |
| *No-implementation* [1] | Imagine that the municipal government has published a waste separation regulation without implementation. Public education is adopted but no supervision or punishment strategies are applied. |
| Policy manipulation check: (1) Will *you/the residents* sort waste according to the recommendation by the government? | |
| Willingness to engage in waste separation behavior: (2) To what extent, how likely were *you/the residents* willing to engage in waste separation behavior according to the sorting guidelines? Participants responded on a 0 to 100 percent scale (0 = not at all, 100 = to a large extent). | |
| Attitude towards waste separation (*self–others* version) (1) *You/the residents* agree that waste separation is important. (2) *You/the residents* agree that waste separation is valuable. (3) *You/the residents* support waste separation policies. (4) Waste separation is relevant to *you/the residents*. (5) *You/the residents* are concerned about waste separation. | |

[1] The condition was only used for Study 2. The words in italic indicate the experimental conditions.

3.1.2. Participants and Procedure

The final sample includes data from 240 participants (137 females; mean age 36.49 ± 14.08 years). The sample characteristics are shown in Table 3. The sample profile in this study is generally similar to the population profile of a census. The sample size estimated by G*Power (Faul, F., Erdfelder, E., Lang, A.-G., & Buchner, A. (Heinrich, Germany)) (Version 3.1.9.2) [64] shows that studying at least 128 participants would afford 80% power to detect a medium effect (*Cohen's f* = 0.25). We further increased the sample size to 243 to adequately detect potential interactions in this study.

Participants were randomly recruited from shopping malls with the highest foot traffic that were considered easily accessible by public transportation [65]. They answered the survey with pen and paper. To enhance internal validity, a pilot study of 20 participants was conducted in Beijing. Some wording in the questionnaire was then refined according to the pilot results to improve the questionnaire's items and structure.

This study was reviewed and approved by the Institutional Review Board of the Institute of Psychology, Chinese Academy of Sciences (protocol code: H21078). Data were collected from 1 December 2019 to 20 January 2020 using a standardized questionnaire during the face-to-face surveys. Three responses were excluded due to incomplete answers. Participants completed the survey in exchange for RMB 5.

**Table 3.** Sample Demographics (Study 1).

| Variables | *N* | Percentage | Census [a] |
|---|---|---|---|
| *Gender* | | | |
| Female | 137 | 57.08% | 48.91% |
| Male | 102 | 42.50% | 51.09% |
| No response | 1 | 0.42% | |
| *Age* | | | |
| Under 20 | 44 | 18.33% | 21.89% |
| 20–29 | 32 | 13.33% | 13.12% |
| 30–39 | 68 | 28.33% | 15.73% |
| 40–49 | 58 | 24.17% | 15.82% |
| 50–59 | 16 | 6.67% | 15.31% |
| 60 or above | 19 | 7.92% | 18.13% |
| No response | 3 | 1.25% | |
| *Education level* | | | |
| Junior high school or below | 24 | 10.00% | 67.68% |
| Special (or technical) secondary school | 15 | 6.25% | 4.73% |
| Senior high school | 34 | 14.17% | 13.01% |
| Junior college | 45 | 18.75% | 7.67% |
| Bachelor's degree | 110 | 45.83% | 6.27% |
| Master's degree | 9 | 3.75% | 0.64% |
| Doctorate | 3 | 1.25% | |
| *Occupation (Working Place)* | | | |
| School | 59 | 24.58% | |
| Company | 81 | 33.75% | |
| Property management company | 9 | 3.75% | |
| Community committee | 13 | 5.42% | |
| At home | 25 | 10.42% | |
| Government office | 22 | 9.17% | |
| Others | 30 | 12.50% | |
| No response | 1 | 0.42% | |

[a] China Statistical Yearbook 2020 [66].

### 3.1.3. Measurement

**Willingness to engage in waste separation behavior.** We examined the policy factor using a scenario in which participants were asked to what extent they or the residents would engage in waste separation behavior according to local government sorting guidelines (see Table 3). Participants responded on a 0 to 100 percent scale (0 = not at all, 100 = to a large extent). A manipulation check was implemented with the question "Will you/the residents sort waste separately according to the requirements?" (1 = participation, 0 = otherwise).

**Attitudes toward waste separation.** Attitudes toward waste separation [67–70] were assessed with five items on 7-point scales ranging from 1 (strongly disagree) to 7 (strongly agree). The items for the *others condition* were identical to those used for the *self condition* except that the word *you* included in each item was replaced with the word *the residents*. The Cronbach's α was 0.87. The overall attitude score was computed as the mean of the

five items. Finally, we collected demographic information (such as age, gender, education level, and occupation).

### 3.1.4. Data Analysis

Microsoft Excel (Redmond, WA, USA) and IBM SPSS Statistics 21.0 IBM (Armonk, NY, USA) software were used for data analysis. The data were pooled in Microsoft Excel after excluding three responses that were incomplete answers. Descriptive statistics were used to describe the participants' characteristics (means ± standard deviations and percentage values). Then, Cronbach's α was calculated to measure the reliability of attitudes towards waste separation. A Chi-squared test was used to conduct a manipulation check. A 2 × 2 between-participants analysis of variance (ANOVA) with demographic variables as covariates was conducted on the participants' willingness to engage in waste separation behavior, and a one-way ANOVA was conducted to compare attitudes toward waste separation in different conditions.

### *3.2. Results*

### 3.2.1. Policy Manipulation Check

A 2 (policy: *mandatory* vs. *voluntary*) × 2 (participation: *yes* vs. *otherwise*) crosstab Chi-squared test was used to test the validity of the manipulation check. 76.58% of the participants in the *mandatory condition* (*n* = 111) reported they would sort waste according to the recommendation by the government, but only 46.51% of the participants in the *voluntary condition* (*n* = 129) reported that they would do so. The difference was significant, $\chi^2$ (1, *n* = 240) = 22.55, *p* < 0.01. These results indicated that the variations in the respondents' responses were purely based on the treatment variations. For brevity, we did not report the manipulation check results in Studies 2 and 3.

### 3.2.2. Willingness to Engage in Waste Separation Behavior

In order to test the effects of policies from the perspective of self–others discrepancies, we conducted a 2 (policy: *mandatory* vs. *voluntary*) × 2 (person: *self* vs. *others*) ANOVA on the participants' willingness to engage in waste separation behavior, and the demographic variables were added in the analysis as covariates. Gender was coded "1" as "female" and "2" as "male". Education level was coded from 1 to 7 as "Junior high school or below" to "Doctorate". Occupation was coded from 1 to 7 as "School" to "Others" (see Table 3). Age was analyzed as a continuous variable.

The results are summarized in Figure 1. The main effect for the policy was significant, $F(1,232) = 9.29$, *p* = 0.003, $\eta_p^2 = 0.04$, and 95% *CI* [3.12,14.52], and participants predicted having a greater willingness to engage in waste separation behavior in the *mandatory condition* (*n* = 111, *M* = 70.36%, *SD* = 25.14%) than in the *voluntary condition* (*n* = 129, *M* = 59.16%, *SD* = 25.40%). The interaction between the policy and the person conditions was not significant, $F(1,232) < 1$. There was a significant main effect of the person, $F(1,232) = 65.67$, *p* < 0.001, $\eta_p^2 = 0.22$, and 95% *CI* [18.02,29.59], as participants predicted themselves (*n* = 107, *M* = 78.29%, *SD* = 20.31%) having a greater willingness to separate waste than others (*n* = 133, *M* = 53.12%, *SD* = 24.35%). These findings are consistent with Hypothesis 2. We observed robust self-enhancement bias, which involves taking a favorable view of oneself. Demographic variables, as covariates, had no significant effect on the willingness to engage in waste separation behavior except age (see Table 4). We did not consider these demographic variables in Studies 2 and 3. We examine the age effect in Study 3.

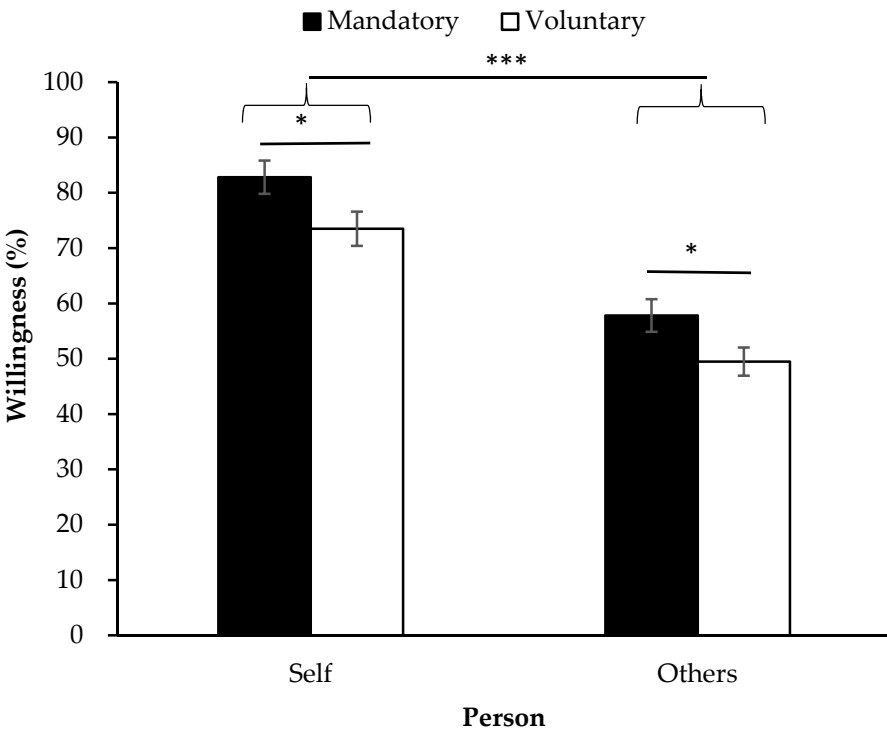

**Figure 1.** Willingness to engage in waste separation behavior perceived by *self* and *others* for two policy conditions applied in Study 1. Note. The error bars indicate standard error. * $p < 0.05$, *** $p < 0.001$.

**Table 4.** Tests of Between-Subjects Effects in Study 1.

| | | | | | | |
|---|---|---|---|---|---|---|
| | **Dependent Variable: Willingness to Sort Wastes** | | | | | |
| **Source** | **Type III Sum of Squares** | *df* | **Mean Square** | *F* | **Sig.** | **Partial Eta Squared** |
| Corrected model | 46,301.393 [a] | 7 | 6614.485 | 13.547 | 0.000 | 0.290 |
| Intercept | 33,784.866 | 1 | 33,784.866 | 69.194 | 0.000 | 0.230 |
| Gender | 97.510 | 1 | 97.510 | 0.200 | 0.655 | 0.001 |
| Education level | 432.659 | 1 | 432.659 | 0.886 | 0.348 | 0.004 |
| Occupation | 1551.415 | 1 | 1551.415 | 3.177 | 0.076 | 0.014 |
| Age | 2776.677 | 1 | 2776.677 | 5.687 | 0.018 | 0.024 |
| Person | 32,062.880 | 1 | 32,062.880 | 65.667 | 0.000 | 0.221 |
| Policy | 4535.388 | 1 | 4535.388 | 9.289 | 0.003 | 0.038 |
| Person × Policy | 1.840 | 1 | 1.840 | 0.004 | 0.951 | 0.000 |
| Error | 113,276.898 | 232 | 488.262 | | | |
| Total | 1,153,098.556 | 240 | | | | |
| Corrected total | 159,578.291 | 239 | | | | |

[a] R Squared = 0.290 (Adjusted R Squared = 0.269).

### 3.2.3. Attitude toward Waste Separation

A one-way ANOVA was conducted to compare attitudes toward waste separation in the *self* and *others* conditions. There was a significant self–others discrepancy (see Figure 2), $F(1,238) = 33.17$, $p < 0.001$, $\eta_p^2 = 0.12$, 95% *CI* [0.50,1.02], and participants perceived themselves ($n = 107$, $M = 6.02$, $SD = 0.82$) to have a more positive attitude toward waste separation than others ($n = 133$, $M = 5.26$, $SD = 1.14$). These results reveal strong self-enhancement bias in attitudes toward waste separation.

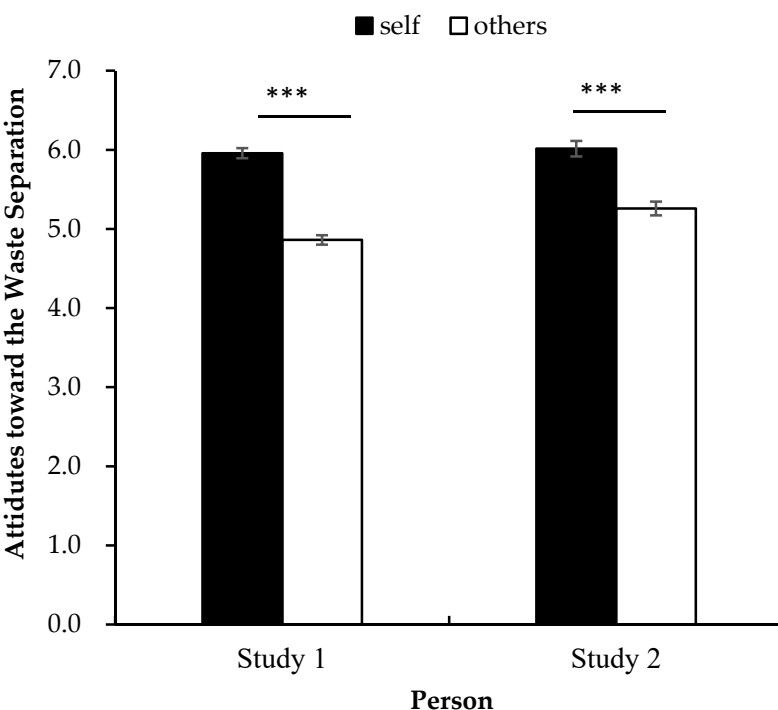

**Figure 2.** Attitudes toward waste separation for the *self* and others conditions for Studies 1 and 2. Note. The error bars indicate standard error. *** $p < 0.001$.

*3.3. Discussion*

Study 1 demonstrates that a mandatory policy would increase the likelihood of waste separation behavior. Mandatory policies are an effective means to facilitate residents' willingness to engage in waste separation. We also observed a robust self-enhancement bias. Specifically, residents perceived themselves to be better than others in both their willingness to adopt and attitudes toward waste separation behavior. However, Study 1 did not design a baseline condition in which there was a no-implementation policy related to waste separation. To replicate and extend the findings of Study 1, we performed Study 2 by adding a baseline condition (no-implementation policy).

**4. Study 2**

In order to confirm the findings of Study 1, we conducted Study 2 with four variations. First, data were collected online due to the COVID-19 pandemic from 1 to 10 February 2020. Second, participants assessed their willingness to engage in waste separation behavior on 11-point scales ranging from 0 to 10. Third, there were three policy conditions (*mandatory* vs. *voluntary* vs. *no-implementation*, see Table 2). Finally, we collected self-reported waste separation behavior in participants' real lives to provide converging evidence for our hypothesis.

*4.1. Materials and Methods*

4.1.1. Research Design

The study employed a 3 (policy: *mandatory* vs. *voluntary* vs. *no-implementation*) × 2 (person: *self* vs. *others*) between-participants design. Due to this design, each participant randomly received one of the six versions of the questionnaire and completed it individually.

4.1.2. Participants

The participants were randomly recruited from several cities, including Beijing, Shanghai, Suzhou, Shangrao, and others. The final sample includes data from 349 participants (229 females; mean age 33.20 ± 9.63 years). The sample characteristics are shown in Table 5.

The sample size estimated by G*Power 3.1 indicates that involving at least 158 participants would afford 80% power to detect a medium effect (*Cohen's f* = 0.25). We further increased the sample size to 359 to adequately detect potential interactions in the current study. We discarded data from 10 participants who gave the same rating for all items.

**Table 5.** Sample Demographics (Study 2).

| Variables | *n* | Percentage |
|---|---|---|
| *Gender* | | |
| Female | 229 | 65.62% |
| Male | 120 | 34.38% |
| *Age* | | |
| Under 20 | 41 | 11.75% |
| 20–29 | 56 | 16.05% |
| 30–39 | 174 | 49.86% |
| 40–49 | 52 | 14.90% |
| 50–59 | 19 | 5.44% |
| 60 or above | 3 | 0.86% |
| No response | 4 | 1.15% |
| *Education level* | | |
| Primary school | 3 | 0.86% |
| Junior high school | 20 | 5.73% |
| Senior high school | 34 | 9.74% |
| Junior college | 24 | 6.88% |
| Bachelor's degree | 133 | 38.11% |
| Master's degree | 84 | 24.07% |
| Doctorate | 51 | 14.61% |
| *Occupation (Working place)* | | |
| Company | 96 | 27.51% |
| School or research institute | 155 | 44.41% |
| Government office | 10 | 2.87% |
| Public welfare social organizations | 4 | 1.15% |
| Liberal professions | 32 | 9.17% |
| At home | 8 | 2.29% |
| Others | 44 | 12.61% |
| *Local policy* | | |
| Mandatory | 103 | 29.51% |
| Voluntary | 181 | 51.86% |
| Uncertain | 65 | 18.62% |

*Note.* Local policy category is obtained from the question "Is there a mandatory waste separation policy in your city?".

### 4.1.3. Measurement

Participants' self-reported waste separation behavior in real life was measured with six items on 5-point scales ranging from 1 (never) to 5 (always) [23,71,72]. The Cronbach's $\alpha$ was 0.74. The other materials used were the same as those used in Study 1. A pilot test with 10 respondents was conducted to refine some wording in the questionnaire. In accordance with the respondents' feedback, we refined certain expressions before creating the final online questionnaire.

The six items of self-reported waste separation behavior are as follows: (1) I sort out food waste at home; (2) I sort out battery and electronic waste at home; (3) I reuse plastic bags; (4) I sort out recyclable materials (i.e., plastic bottles, paper, and cans) at home; (5) I dispose of waste strictly in accordance with the classification icons; (6) I mix all the waste together.

### 4.1.4. Data Analysis

The data analysis process was the same as the process in Study 1.

*4.2. Results*

4.2.1. Willingness to Engage in Waste Separation Behavior

A 2 (person: *self* vs. *others*) × 3 (policy: *mandatory* vs. *voluntary* vs. *no-implementation*) ANOVA was performed to examine the effects of policies and person conditions on the willingness to separate waste (see Figure 3). Consistent with the self-enhancement bias found in Study 1, the analysis reveals a significant main effect for the person, $F(1,343) = 66.82$, $p < 0.001$, $\eta_p^2 = 0.16$. Participants perceived themselves ($n = 166$, $M = 8.03$, $SD = 1.97$) to have a greater willingness to engage in separation behavior than others ($n = 183$, $M = 6.09$, $SD = 2.17$). Furthermore, a significant main effect was found for the policy, $F(2,343) = 18.05$, $p < 0.001$, $\eta_p^2 = 0.10$, indicating that a mandatory policy ($n = 87$, $M = 8.11$, $SD = 1.67$) is associated with a greater willingness to engage in separation behavior rather than a voluntary policy ($n = 143$, $M = 6.86$, $SD = 2.15$) and no-implementation policy ($n = 119$, $M = 6.40$, $SD = 2.57$). The difference between the *voluntary condition* and *no-implementation policy condition* was not significant.

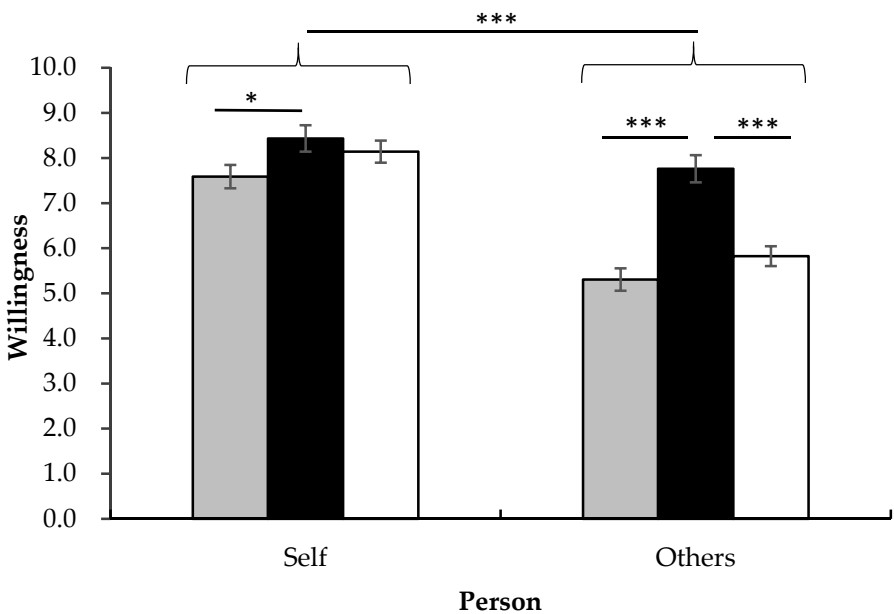

**Figure 3.** Willingness to engage in waste separation behavior perceived by the *self* and *others* under the three policy conditions of Study 2. Note. The error bars indicate standard error. * $p < 0.05$, *** $p < 0.001$.

The interaction between the person and policy conditions was significant, $F(2,343) = 5.62$, $p = 0.004$, $\eta_p^2 = 0.032$. Further analyses indicated that when a participant predicted others' willingness to engage in waste separation behavior, the difference between the three policy conditions was significant, $F(2,343) = 20.93$, $p < 0.001$, $\eta_p^2 = 0.11$. In line with Hypotheses 1a and 1b, participants predicted others to show a greater willingness to separate waste in the *mandatory condition* ($n = 42$, $M = 7.76$, $SD = 1.61$) than in the *no-implementation policy condition* ($n = 62$, $M = 5.31$, $SD = 2.26$), $p < 0.001$, 95% *CI* [1.68,3.23] and the *voluntary condition* ($n = 79$, $M = 5.82$, $SD = 1.89$), $p < 0.001$, 95% *CI* [1.20,2.68]. There was no significant difference between the *no-implementation policy condition* and the *voluntary condition*, $p = 0.12$. These results indicate that voluntary or no-implementation policies cannot facilitate waste separation.

When participants predicted their own willingness to engage in separation behavior, we found an insignificant effect of policy conditions, $F(2,343) = 2.50$, $p = 0.084$, $\eta_p^2 = 0.014$. The difference between the *mandatory* ($n = 45$, $M = 8.43$, $SD = 1.68$) and the *voluntary* ($n = 64$, $M = 8.14$, $SD = 1.72$) conditions failed to reach significance, $p = 0.44$. These results indicate

that people perceived themselves to perform waste separation well in both the *mandatory* and *voluntary conditions*.

### 4.2.2. Self-Reported Waste Separation Behavior

To examine residents' waste separation behavior in real lives, we performed a one-way ANOVA using local policy as the independent variable. The local policy category was obtained from the question: Is there mandatory waste separation in your city? The effect of local policies was significant, $F(2,346) = 17.62$, $p < 0.001$, $\eta_p^2 = 0.09$. Participants reported more waste separation behavior when a mandatory policy ($n = 103$, $M = 2.98$, $SD = 0.62$) was adopted in their city than under voluntary ($n = 181$, $M = 2.53$, $SD = 0.64$), $p < 0.001$, 95% *CI* [0.27,0.64] and uncertain conditions ($n = 65$, $M = 2.65$, $SD = 0.56$), $p = 0.003$, 95% *CI* [0.09,0.56]. The results provide further support for the view that mandatory policies facilitate waste separation.

### 4.2.3. Attitudes toward Waste Separation

To examine attitudes toward waste separation in the two person conditions (see Figure 2), a one-way ANOVA was conducted. The effect for *the person* was found to be significant, $F(1,347) = 159.07$, $p < 0.001$, $\eta_p^2 = 0.31$, 95% *CI* [0.93,1.27]. Consistent with the results on self-enhancement bias of Study 1, participants perceived themselves ($n = 166$, $M = 5.96$, $SD = 0.73$) to have a more positive attitude toward waste separation than others ($n = 183$, $M = 4.86$, $SD = 0.88$).

### *4.3. Discussion*

Study 2 provides a further understanding of the effect of mandatory policies on waste separation behavior perceived by the self and others. First, we replicate the robust self-enhancement bias whereby people perceive themselves to engage in more waste separation behavior and to have a more positive attitude toward waste separation than others, whether under a mandatory policy or voluntary policy. If everyone performed waste separation well under a voluntary policy, public participation should be high. However, both a voluntary policy and no-implementation policy are ineffective in encouraging waste separation in a natural setting.

Second, we find that a mandatory policy with supervision can actually facilitate waste separation behavior. The participants reported being more willing to engage in waste separation behavior and engaged in more waste separation behavior under the mandatory policies than the other two conditions. Surprisingly, participants' own willingness to engage in waste separation behavior appeared to be similar under the voluntary policy and mandatory policy conditions. However, participants reported that others' willingness to separate waste would be promoted by a mandatory policy rather than by a voluntary policy. This self-enhancement bias became salient from the anonymous questionnaire administered online. As the participants of Studies 1 and 2 were adults, it was unclear whether self-enhancement bias could be observed for adolescents. As noted above in the section of the literature review, adolescents play key roles in protecting the natural environment [58–61]. Thus, we performed Study 3.

## 5. Study 3

Study 3 was a replication of Study 2 with four variations. First, we adopted eight items for waste separation behaviors (see Appendix A) that provided high reliability in the hypothetical scenarios. Second, the person variable (*self* vs. *others*) was manipulated within participants to confirm the self-enhancement bias. Third, the participants were junior high school students to test the generality of self–others discrepancies. Finally, the *no-implementation policy condition* was excluded due to its insignificant effect compared to that found under the *voluntary condition* in Study 2.

### 5.1. Materials and Method

5.1.1. Research Design, Measurement, and Procedures

This study utilized a 2 (policy: *mandatory* vs. *voluntary*) × 2 (person: *self* vs. *others*) mixed design where the policy condition applied a between-participants design and the person condition applied a within-participants design. For each condition, participants measured waste separation behavior with eight items (see Appendix A) on 11-point scales ranging from 0 (not at all) to 10 (to a large extent) [71,72]. The Cronbach's α was 0.86.

Participants were randomly assigned to the four conditions (*self-mandatory* vs. *self-voluntary* vs. *others-mandatory* vs. *others-voluntary*) in September 2020. Half of the participants completed the *self condition* first and completed the *others condition* the next day. The other half of the participants completed the *others condition* first and completed the *self condition* the next day.

5.1.2. Participants

The study was conducted in a junior high school located in Dongguan, a city in southern China. We recruited 20 students for the pilot study, and another 196 students participated in the formal study. The sample size estimated by G*Power 3.1 indicated that studying at least 66 participants would afford 80% of power to detect a medium effect (*Cohen's f* = 0.25). Participants who gave the same answers on all behavior items or incomplete answers were excluded. The final sample includes data from 121 participants (65 females; mean age 13.14 ± 0.45 years).

5.1.3. Data Analysis

The data analysis process was the same as the process in Study 1.

### 5.2. Results

A 2 (policy: *mandatory* vs. *voluntary*) × 2 (person: *self* vs. *others*) repeated-measures ANOVA was performed using the willingness of waste separation behavior as the dependent variable (see Figure 4). Consistent with the results of Study 2, the interaction between the person and the policy was significant, $F(1,119) = 5.63$, $p = 0.019$, $\eta_p^2 = 0.045$. In the *others condition*, students predicted others' greater willingness to engage in waste separation behavior in the *mandatory condition* ($n = 62$, $M = 5.86$, $SD = 1.70$) was significant more than that in the *voluntary condition* ($n = 59$, $M = 5.19$, $SD = 1.60$), $p = 0.028$, 95% *CI* [0.08,1.27]. However, in the *self condition*, the difference in willingness to separate waste between the *mandatory* and *voluntary conditions* failed to reach significance, $p = 0.95$. No significant main effect of the policy was found, $F(1,119) = 1.70$, $p = 0.20$, $\eta_p^2 = 0.01$. The analysis reveals a significant main effect for the person condition, $F(1,119) = 73.54$, $p < 0.001$, $\eta_p^2 = 0.38$, 95% *CI* [0.91,1.46]. This result suggests that students predicted themselves ($n = 121$, $M = 6.71$, $SD = 1.61$) to have a greater willingness to engage in separation behavior than others ($n = 121$, $M = 5.53$, $SD = 1.68$).

### 5.3. Discussion

These data provide further evidence for the robustness of self-enhancement bias, which was also found for the students in junior high school. All three studies indicate that people always assess themselves to be better than others in waste separation behavior. Students tended to perceive themselves as doing as well in both the *mandatory* and *voluntary conditions*, but mandatory policies facilitated others' waste separation behavior more efficiently.

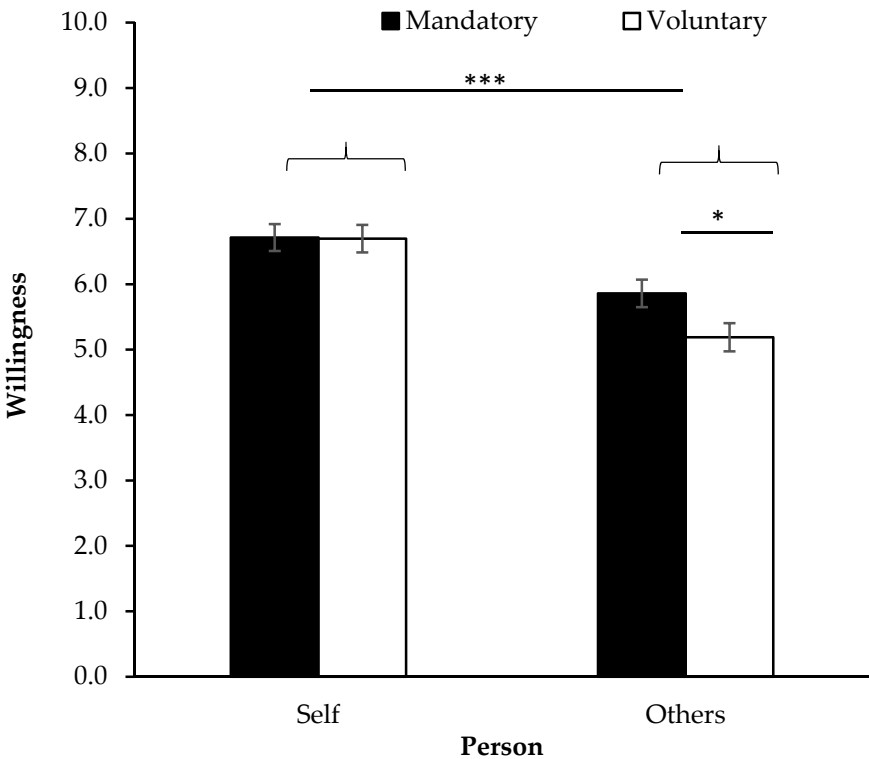

**Figure 4.** Willingness to engage in waste separation behavior perceived for *self* and *others* for the two policy conditions adopted in Study 3. Note. The error bars indicate standard error. * $p < 0.05$, *** $p < 0.001$.

## 6. General Discussion

Waste separation is a significant challenge and has become a social dilemma globally. We attempted to solve this dilemma by investigating the effect of mandatory policies on waste separation from the perspective of the self versus others in the present research. All of our studies consistently demonstrate that a mandatory policy with supervision facilitates waste separation behavior. However, a policy without supervision (voluntary policy or no-implementation policy) does not facilitate waste separation behavior. We also observed a robust self-enhancement bias. Participants estimated themselves to engage in more waste separation behavior and to have more positive attitudes toward waste separation than others. Consistent with prior studies on mandatory policies [7,10,13,18,20,36], we highlight the impact of a mandatory policy with supervision in reducing contamination in waste separation, which has implications for policymakers to design waste management strategies to achieve zero-waste goals.

### 6.1. Mandatory Waste Separation Policies

Our three studies consistently demonstrate that a mandatory policy with supervision is an efficient means to facilitate waste separation behavior. It is consistent with previous studies [6,7,46], although few studies addressed the role of supervision in waste separation. We further found that supervision or government intervention is an essential tool to make policies effective. A policy without supervision is as ineffective at encouraging waste separation as no policy. The finding is consistent with the study conducted by Hao et al. [24,73], which found the waste separation rate was still low even under a policy due to a lack of clear supervision, intervention, or punishment. The deterrence effect could be absent without supervision due to a lack of perceived punishment celerity, certainty, and severity based on deterrence theory [20,42]. For instance, there are specially-assigned persons responsible for supervision beside garbage bins in Shanghai. If you mix waste, they will guide you in the correct separation or reject the waste and even fine you [40]. Thus,

promoting waste separation requires not only mandatory policies but also supervision and punishment such as specially-assigned persons for guidance and fine penalties to regulate residents' behaviors.

### 6.2. Self-Enhancement Bias

We observed universal self-enhancement bias in the domain of waste separation in three studies. Our findings are consistent with previous studies conducted in other domains such as personality traits, abilities, and green behaviors [52,53,57]. Our participants perceived themselves to be better than others in both their willingness to perform and attitudes toward waste separation behavior. Although people perceived themselves to perform as well under both the *mandatory* and *voluntary conditions* (self-enhancement bias), a mandatory policy with supervision plays a key role in promoting waste separation behavior in real life. Sometimes, people value introspection on their own intentions when evaluating themselves but only value behavior when evaluating others [74]. Thus, people assessed themselves according to their support for waste separation, while they assessed others according to actual waste separation behavior.

In China, most residents support a waste separation policy, but few people sort waste on a daily basis [22,75]. Self–others discrepancies might decrease people's waste separation behavior and cooperation. Bergquist [57] proposed a similar view that self-enhancement bias likely reduces people's sense of obligation and pro-environmental behavioral intentions, which might create a barrier to future pro-environmental behavior. It is possible that feedback, prompt, information publicity, public education, and others might reduce self-enhancement bias in the domain of waste separation. Nevertheless, there is a lack of empirical evidence. Thus, to promote waste separation behavior, future research must explore ways to reduce self-enhancement bias.

### 6.3. Adolescent Participants

The participants in our research were recruited from different age groups, and Study 3 focused on adolescents. Our findings are in line with previous research advocating that adolescents predict themselves to have a greater willingness to engage in separation behavior [58–61]. Due to adolescents adapting to new trends in society more easily and being able to actively influence their parents in particular [59], we may conclude that the implementation of waste separation among adolescents primarily can be an effective strategy for promoting waste separation behavior on a national level. Moreover, it is easier to promote mandatory waste separation and environmental education programs in such a special context of schools. To sum up, according to our positive results, we recommend that policymakers promote waste separation programs among adolescents in schools.

### 6.4. Policy Recommendations

First, mandatory policies as an effective tool to facilitate waste separation are necessary to regulate residents' waste separation behaviors. Second, supervision, such as with specially-assigned persons for guidance or monitoring, is important to ensure the implementation of various policies. A paid job is set to supervise residents' waste separation behaviors based on regulations and guidelines. The specially-assigned person can guide residents to separate correctly and return the mixed waste. For example, in Japan and Sweden, there are mandatory policies with actual supervision that have led to the long-term effectiveness of waste separation by people. Finally, publicity and education should make people pay more attention to their behavior objectively in order to reduce self-enhancement bias. The use of supervision, commitment, information publicity, and feedback could be effective methods to make people focus on their actual behaviors instead of intentions.

### 6.5. Limitations and Future Research

Our research highlights the key roles of mandatory policies with supervision in facilitating waste separation behavior and self-enhancement bias. However, there are several limitations that require further investigation. First, the limitations of mandatory policies should be addressed. For instance, a mandatory policy may be ineffective for those with a less pro-environmental worldview [20,76]. Enforcement may also crowd out intrinsic motivation [62] and reduce the positive effects of voluntary participation, which needs further investigation. Second, self-reported results might contain elements of desirability bias or overestimation, which need to be treated with caution [77]. Many people would "exaggerate" their waste separation behavior due to the "intention–behavior gap" [25,26]. Actual behavior needs further investigation through empirical studies. Third, there was a significant age-related difference in the willingness to engage in waste separation behavior. The effect of age and reduction of self-enhancement bias need further investigation. Finally, we recruited different participants in various situations (shopping mall and school) and with different roles (residents and adolescents). Further studies must test the findings of our studies in other contexts, such as workplaces or communities.

## 7. Conclusions

The present research provides empirical evidence for the effectiveness of a mandatory policy with supervision and proves the key role of supervision in policy implementation. We also observed robust self-enhancement bias in waste separation and extended the research on self-improvement bias to the pro-environment field. Our studies provide a deeper understanding that can promote engagement in waste separation behavior. Although people perceive themselves to be better than others in both their willingness to perform and attitudes toward waste separation behavior, residents' waste separation behavior should be monitored to enhance social norms. Some policy recommendations are proposed for policymakers to improve the effectiveness of policies. It is necessary to increase waste separation quality by recruiting volunteers and clarifying management responsibilities. A mandatory policy with actual supervision is necessary to ensure the long-term effectiveness of waste separation.

**Author Contributions:** P.L. conceived and designed the studies 1–2; P.L., Y.Z. and R.C. conceived and designed the Study 3. Formal analysis, Y.Z., R.C. and P.L.; resources, Y.Z., R.C. and P.L.; data curation, P.L.; writing—original draft preparation, Y.Z., R.C. and P.L.; writing—review and editing, Y.Z., R.C., P.L., M.Y. and B.H.; supervision, P.L. All authors have read and agreed to the published version of the manuscript.

**Funding:** This research was partly funded by the Scientific Foundation of Institute of Psychology, Chinese Academy of Sciences (No. Y9CX391008), the joint program of Chinese Academy of Sciences and Japan Society for the Promotion of Science (No. GJHZ2095), and the National Natural Science Foundation of China (No. 72174194).

**Institutional Review Board Statement:** The study was conducted according to the guidelines of the Declaration of Helsinki, and approved by the Institutional Review Board of the Institute of Psychology, Chinese Academy of Sciences (protocol code: H21078; date of approval: 15 March 2021).

**Informed Consent Statement:** Informed consent was obtained from all subjects involved in the study.

**Data Availability Statement:** If reader have questions about this paper or would like more data information, please feel free to contact the corresponding author. We will provide the raw data.

**Acknowledgments:** We are grateful to all the participants in this research.

**Conflicts of Interest:** The authors declare no conflict of interest.

## Appendix A

**Table A1.** Design and measurement in Study 3.

| Conditions (*Self–others* version for the hypothetical scenario) | |
| --- | --- |
| Mandatory | Imagine that our school has implemented a waste separation policy and set up sortable garbage bins. There is not only public education but also a specially assigned student that supervises waste separation in each class. |
| Voluntary | Imagine that our school has implemented a waste separation policy and set up sortable garbage bins. There is only public education and no supervision or punishment strategies adopted in each class. |
| **Willingness of waste separation behavior** | |
| How likely are *you/other* students willing to:<br>(1) Engage in waste separation behavior?<br>(2) Put empty plastic bottles into the recycling bin?<br>(3) Throw unfinished bread into the kitchen waste bin?<br>(4) Put used dirty paper into the other trash bin?<br>(5) Put empty cans into the recycling bin?<br>(6) Throw an unfinished steamed stuffed bun into the kitchen waste bin?<br>(7) Put dirty food packaging bag into the other trash bin?<br>(8) Empty plastic water bottles before throwing them into the recycling bin? | |

*Note.* The words in italic indicate the experimental conditions.

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
