# Peer review of "I Am Better Than Others: Waste Management Policies and Self-Enhancement Bias"

_sustainability, doi:10.3390/su132313257_

Round 1

Reviewer 1 Report

Overall comments:

In general, the authors were able to show an overview of the need to impose mandatory policy for waste separation management, with theory, literature, and real examples supporting. However, throughout the presentation of the manuscript, especially the literature part, the distinguishment between mandatory policy without government intervention and volunteer policy was not clearly made. Details comments are as follows:

  1. Abstract 

Please separate the methods and results of the 3 studies, instead of just saying (Study 1-3) 

    2. Introduction 

Your introduction has sufficiently illustrated the overall construct of your manuscript, however the flow starting from line 49 need to be improved. 

In line 49, you start with stating the lack of public participation in waste source separation, however, your next sentences are the recycling rates of other countries regardless of the publication participation. I suggest the flow of your paragraph from line 49-line 66 will be :

a. Starting with the possibility of the affection due to different regional or states' policy, therefore leading to the illustration of the different recycling rates range in USA and UK (that you have stated in 50-54). 

b. After that, insert some more global/international waste separation policy of those countries where did a great job in waste separation management, supporting with  more other statistics of course. 

c. This show policy would affect the effectiveness of the waste management. Moreover, stick with your original paragraph (Line 69-66), stating the situation and examples of China. (Additionally suggestions: find more recent examples yet not 20 years ago.) 

d. Bring back the paragraph in Line 54, stating, although there are strategies implemented, public participation remains low..... ; so this brings you forward getting the importance of public participation into your research focus. 

Line 81-85 should be adjusted. I suggest you to put a new subtitle as literature review for part 1.1 and 1.2. It's not that academically structure by stating "This paper is organized as follows". Separating the Introduction session and Literature Review session will automatically make the organization clear. 

Part 1.1 Line 88 "Although most ......correlational". I think this sentence can be deleted. 

Please be alert towards your use of tense, should be consistent throughout the manuscript, I prefer using past tense and present perfect tense when stating the literature that you have reviewed. 

Starting from line 110 -141, although I could barely understand the points and arguments that you have pointed out, however, you did not state clearly that there should be 3 types of policy which you have or/and you are going to involve in your research measurement. 

The 3 types are 1. Mandatory policy  with supervision or government intervention 2. Mandatory policy without supervision or government intervention 3. Volunteer policy

However, throughout the paragraph, the wordings and illustrations showed that you might have mixed up the differences, or you did not distinguish the differences between the 2nd type and the 3rd type of policy. 

Taking line 123-125 as an example, mandatory policy should be considered as compulsory, while, an ineffective mandatory policy (was not intervened by the government doesn't mean its remains voluntary. Therefore, you could not use the "in other words" this is not considered as explaining the same thing.  Moreover, the sentence in line 123-124 is not academically presented, the word "checked", please rephrase it.

In this paragraph, more literature/example or even citing policy document are needed to show or address examples of different types of policy, including those volunteer policies. 

Part 1.2 

Advance English editing is needed.

Eg. 151-153 ; Line 173-175 

Hypothesis: 

Baseline measure (Mandatory policy without supervision/government intervention) should be stated as one of the hypothesis as well. 

2. Method 

The overall content of the method and results of study 1-3 were well presented. However, All studies have not presented the data analysis before presenting the results. Please add back the data analysis paragraph into each study. 

3. Study 2

Please separate the measurements that you have mentioned in the first paragraph of study 2. Although there were mentioned in Study 1, it should be belonged to the measurement; so please also separate design and measurement of part 3.1.1 .

Line 386-389 should be considered as a discussion interpretation. 

Line 422-427 made me feel so confused. Where did the participants report their willingness in Study 2 but show no significant differences? 

Study 3

You didn't mention any review or information regarding adolescents, so why would you involve high school students in Study 3? 

5. General Discussion 

Line 499, same issue, mandatory policy without supervision is not as same as volunteer policy... 

Line 508-509, the questioning form of illustration is considered as not academically presented. 

More elaboration of your interpretation is needed. E.g. Line 511-512, how did the lack of clear supervision and punishment affect the waste separation rate, what's meant by punishment ? How should a strong and robust supervision be performed? I believe this research is not only a research to show the effect of self-enhancement bias, while the implication of the research towards the Mainland China waste management policy is very significant as well. 

Part 5.2 

More elaboration and suggestions on ways to reduce self-enhancement bias should be addressed as well. 

Part 5.3 

You mentioned the limitation is the lack of pro- environmental worldview, however, I believe that one of the purposes of adopting mandatory policy is to build the pro- environmental worldview of  Mainland China/the place, therefore, the limitation might not be caused by the city itself or the person but the  ineffectiveness of mandatory policy. Hence, you might need to provide some more personal perspective on how to address or reduce this limitation of the city.

Author Response

Response to Reviewer 1 Comments

Point 1:  Abstract- Please separate the methods and results of the 3 studies, instead of just saying (Study 1-3)

Response 1: Thank you for the suggestion! We have revised the Abstract according to the suggestion to introduce more in detail.

Point 2: Introduction 

  1. Starting with the possibility of the affection due to different regional or states' policy, therefore leading to the illustration of the different recycling rates range in USA and UK (that you have stated in 50-54).
  2. After that, insert some more global/international waste separation policy of those countries where did a great job in waste separation management, supporting with more other statistics of course.
  3. This show policy would affect the effectiveness of the waste management. Moreover, stick with your original paragraph (Line 69-66), stating the situation and examples of China. (Additionally suggestions: find more recent examples yet not 20 years ago.)
  4. Bring back the paragraph in Line 54, stating, although there are strategies implemented, public participation remains low..... ; So this brings you forward getting the importance of public participation into your research focus.

Response 2: We appreciate your detailed and very helpful suggestions very much! We have revised the section to be clearer and more logical. The current policy situation in China was also added in this section.

Point 3:  Line 81-85 should be adjusted. I suggest you to put a new subtitle as literature review for part 1.1 and 1.2. It's not that academically structure by stating "This paper is organized as follows". Separating the Introduction session and Literature Review session will automatically make the organization clear. 

Response 3: Thank you for the suggestion! We have separated the Introduction and Literature Review to make the organization clearer.

Point 4:  Part 1.1 Line 88 "Although most ......correlational". I think this sentence can be deleted. 

Response 4: Thank you for the suggestion! Line 88 "Although most ......correlational" is now deleted.

Point 5:  Please be alert towards your use of tense, should be consistent throughout the manuscript, I prefer using past tense and present perfect tense when stating the literature that you have reviewed. 

Response 5:  Thank you very much! We are very sorry for our negligence of the use of tense. They have been revised.

Point 6:  Starting from line 110 -141, did not state clearly that there should be 3 types of policy which you have or/and you are going to involve in your research measurement. 

Taking line 123-125 as an example, you could not use the "in other words" this is not considered as explaining the same thing.  Moreover, the sentence in line 123-124 is not academically presented, the word "checked", please rephrase it.

In this paragraph, more literature/example or even citing policy document are needed to show or address examples of different types of policy, including those volunteer policies.

Response 6:  Thank you for the comment and clarification question! We consulted more literatures and redefined three types of policy (mandatory policy with supervision, voluntary policy without supervision, and no-policy) based on the literature and the policy situation. Related content in Literature Review and Discussion were also revised. Mandatory policy was defined as the policy implemented with supervision, voluntary policy was defined as the policy implemented without supervision, and no policy was defined as the policy without implementation.

Point 7:  Part 1.2 - Advance English editing is needed. Eg. 151-153 ; Line 173-175 

Response 7: Thank you for the suggestion! We have employed American Journal Experts for academic writing editing in English language.

Point 8:  Hypothesis - Baseline measure (Mandatory policy without supervision/government intervention) should be stated as one of the hypothesis as well. 

Response 8:  Thank you for the suggestion! Baseline measure redefined as no policy implemented was added in the hypothesis.

Point 9:   Method - All studies have not presented the data analysis before presenting the results. Please add back the data analysis paragraph into each study. 

Response 9:  Thank you for pointing out the insufficiency of the organization. The data analysis has been added in each study.

Point 10:  Study 2 - Please separate the measurements that you have mentioned in the first paragraph of study 2. Although there were mentioned in Study 1, it should be belonged to the measurement; so please also separate design and measurement of part 3.1.1 .

Response 10: Thank you for your detailed suggestions! We have separated design and measurement in Study 2.

Point 11: Line 386-389 should be considered as a discussion interpretation. 

Response 11: Thank you for your detailed suggestions! Line 386-389, “If everyone ...... in a natural setting.” was put into Discussion. And more detailed discussion was followed.

Point 12: Line 422-427 made me feel so confused. Where did the participants report their willingness in Study 2 but show no significant differences? 

Response 12:  In line 380-383 (Line 572-575 in modified version), there was no significance across the mandatory and voluntary conditions when they predicted their own willingness to engage in waste separation behavior. However, when they predicted others’ willingness to engage in waste separation behavior, there was significant difference between mandatory and voluntary conditions.

Point 13: Study 3 -You didn't mention any review or information regarding adolescents, so why would you involve high school students in Study 3? 

Response 13:  We appreciate your helpful suggestions very much! We added the reason to choose the adolescents as the participants based on some literature (line 312-324) and discussed the results in adolescents (line 778-793).

Point 14:  General Discussion - Line 499, same issue, mandatory policy without supervision is not as same as volunteer policy... 

Response 14:  Thank you for your suggestions! Line 499, the statements of “However, a mandatory policy without supervision (voluntary policy) does not facilitate waste separation behavior.” were corrected as “However, a policy without supervision (voluntary policy) does not facilitate waste separation behavior.”

Point 15:  Line 508-509, the questioning form of illustration is considered as not academically presented. 

Response 15: We are very sorry for our nonacademic writing. Line 508-509, the statements of “How effective are these mandatory policies? Supervision or check is an essential tool.” were revised as “Supervision or government intervention is an essential tool to make policies effective.”

Point 16:  More elaboration of your interpretation is needed. E.g. Line 511-512, how did the lack of clear supervision and punishment affects the waste separation rate, what's meant by punishment? How should a strong and robust supervision be performed? I believe this research is not only a research to show the effect of self-enhancement bias, while the implication of the research towards the Mainland China waste management policy is very significant as well. 

Response 16:  We appreciate your insightful comment and suggestion very much! Policy recommendations were added in the discussion. And we use the example of Shanghai to illustrate the supervision and punishment.

Point 17:  Part 5.2 - More elaboration and suggestions on ways to reduce self-enhancement bias should be addressed as well. 

Response 17:  Thank you for the insightful suggestion! More elaboration and suggestions on ways to reduce self-enhancement bias combined with the policy were added in Policy recommendations.

Point 18:   Part 5.3 - You mentioned the limitation is the lack of pro- environmental worldview, however, I believe that one of the purposes of adopting mandatory policy is to build the pro- environmental worldview of  Mainland China/the place, therefore, the limitation might not be caused by the city itself or the person but the  ineffectiveness of mandatory policy. Hence, you might need to provide some more personal perspective on how to address or reduce this limitation of the city.

Response 18: We appreciate your insightful comment and suggestion very much! We proposed some policy recommendations to improve the effectiveness of policies in the discussion (line 794-805).

Special thanks to you for your good comments. 

Reviewer 2 Report

Dear author/authors,

Overall, I find the manuscript is interesting and well-written. The details of my comments are provided in the file attached.

Author Response

Response to Reviewer 2 Comments

Point 1:  Abstract - Objectives were not stated clearly. Contribution of the study to theory was not clear. Methodology not stated.

Response 1: We appreciate your detailed and very helpful suggestions very much! We have revised the Abstract according to the suggestion to introduce in more detail and clearer.

Point 2: Perhaps, need to update the LR, it would be better to add few more recent LR preferable in the year between 2020-2021.

Response 2: Thank you for the suggestion! We added some literature as follow:

  • Zhao, L., Zou, J., & Zhang, Z. (2020). Does China’s Municipal Solid Waste Source Separation Program Work? Evidence from the Spatial-Two-Stage-Least Squares Models. Sustainability, 12, 1664; doi:10.3390/su12041664.
  • Lin, B., & Guan, C. (2021). Determinants of household food waste reduction intention in China: The role of perceived government control. Journal of Environmental Management, 299, 113577. https://doi.org/10.1016/j.jenvman.2021.113577
  • Zheng, J., Ma, G., Wei, J., Wei, W., He, Y., Jiao, Y., & Han, X. (2020). Evolutionary process of household waste separation behavior based on social networks. Resources, Conservation and Recycling, 161, 105009. https://doi.org/10.1016/j.resconrec.2020.105009
  • Li, W., Jin, Z., Liu, X., Li, G., & Wang, L. (2020). The impact of mandatory policies on residents’ willingness to separate household waste: A moderated mediation model. Journal of Environmental Management, 275, 111226. https://doi.org/10.1016/j.jenvman.2020.111226
  • Žukauskienė, R., Truskauskaitė-Kunevičienė, I., Gabė, V., & Kaniušonytė, G. (2020). "My words matter": The role of adolescents in changing pro-environmental habits in the family. Environment and Behavior, 1, 1-20. https://doi.org/10.1177/0013916520953150
  • Jovarauskaitė, L., Balundė, A., Truskauskaitė-Kunevičienė, I., Kaniušonytė, G., Žukauskienė, R., & Poškus, M. S. (2020). Toward Reducing Adolescents’ Bottled Water Purchasing: From Policy Awareness to Policy-Congruent Behavior. SAGE Open, 10-12, 1–12. https://doi.org/10.1177/2158244020983
  • Xiao, S., Dong, H., Geng, Y., Francisco, M., Pan, H., & Wu, F. (2020). An overview of the municipal solid waste management modes and innovations in Shanghai, China. Environmental Science and Pollution Research, 27:29943–29953. https://doi.org/10.1007/s11356-020-09398-5

Point 3:  Manipulation checks for all the 3 studies need to be explained clearly.

Response 3:  Thank you for the insight suggestion! Manipulation check was implemented with the question “Will you/the residents sort out the waste separately according to the requirements?” And we added the manipulation check at the beginning of the Result (line 416-422).

Point 4:  number of respondents for each cells not been stated clearly.What is the sample size required for each treatment cells?

Response 4: Thank you for the suggestion! This part of the data is supplemented in the results.

Descriptive Statistics (Study 1)

Dependent Variable:   willingness 

person

policy

Mean

Std. Deviation

N

self

mandatory

82.8120

18.44163

55

voluntary

73.5000

21.25706

52

Total

78.2865

20.31075

107

others

mandatory

58.1250

24.95236

56

voluntary

49.4805

23.40412

77

Total

53.1203

24.35471

133

Total

mandatory

70.3573

25.13961

111

voluntary

59.1628

25.40017

129

Total

64.3403

25.83973

240

Descriptive Statistics (Study 2)

Dependent Variable:   willingness

policy

person

Mean

Std. Deviation

N

No-policy

self

7.59

2.364

57

others

5.31

2.264

62

Total

6.40

2.572

119

mandatory

self

8.43

1.681

45

others

7.76

1.605

42

Total

8.11

1.670

87

voluntary

self

8.14

1.715

64

others

5.82

1.893

79

Total

6.86

2.147

143

Total

self

8.03

1.972

166

others

6.09

2.172

183

Total

7.01

2.292

349

Point 5: Why experimental method was chosen need to be justified clearly. Indicate clearly the type of experimental design used. For example, experimental design used was a 2X2 between subject factorial designs. Experimental procedures not clearly explained.

Response 5: We appreciate your detailed and very helpful suggestions very much! We introduce the experimental design in 3.1.1. Design (Line 335-337). And the details of experimental procedures were supplemented in 3.1.3. Participants and procedure such as the pilot and the location selection.

Point 6: Conclusion - The conclusion is too short. Perhaps the authors need to highlight the contributions of this study to theory and practice in the conclusion section.

Response 6:  Thank you for the suggestion! We have made correction according to the comments.

Point 7: References - Perhaps the authors need to cite more recent articles in the year of 2020-2021.

Response 7: Thank you for the suggestion! We added seven recent articles that stated in comment 2.

Special thanks to you for your good comments. 

Reviewer 3 Report

This topic quite interesting, which focus on waste management. However, as an author, you should be followed the standard format of abstract by indicating (e.g., brief intro and significance, what the issue, gap, research objective, materials and methods, results, conclusion and policy implication, consist of theoretical and practical).

Also, I notice that the structure of introduction to end of section not systematically written. You can restructure the section and looking at the flow of every paragraph (to ensure the clear scenario of the topic). Please add, your own arguments and novelty (state of art).

Methodology - Please elaborates comprehensively this section by the following: 

Research design

Population and Sample

Data collection and sampling technique

Instruments (Items)

Data analysis and what tools

Results - Revise the structure of this section, you can focus on the proposed objectives. Report it in this section no need combine it with discussion.... Please, report the results and next section is discussion part.

XX Mandatory waste separation policies

XX Self-enhancement bias

Discussion - you should be stated what the findings of your research and find out similar or not similar (contrast) previous studies findings, as much as you have.

Conclusion - Can be improve on the basis of research objectives. Also, you required to add section limitation & future research and policy implication e.g., theoretical and practical implications.      

Author Response

Response to Reviewer 3 Comments

Point 1: should be followed the standard format of abstract by indicating (e.g., brief intro and significance, what the issue, gap, research objective, materials and methods, results, conclusion and policy implication, consist of theoretical and practical).

Response 1: Thank you for the detailed suggestions! We have re-written this part according to the suggestions.

Point 2: the structure of introduction to end of section not systematically written. You can restructure the section and looking at the flow of every paragraph (to ensure the clear scenario of the topic). Please add, your own arguments and novelty (state of art).

Response 2: Thank you for the suggestions! We divided the Introduction into two section (Introduction and Literature review). And We have revised the introduction to be clearer and more logical (line 70-106).

Point 3: Methodology - Please elaborates comprehensively this section by the following: Research design, Population and Sample, Data collection and sampling technique, Instruments (Items), Data analysis and what tools

Response 3: Thank you for the detailed suggestions! We have tried our best to improve the structure and made some changes in the manuscript.

Point 4:  Results - Revise the structure of this section, you can focus on the proposed objectives. Report it in this section no need combine it with discussion.... Please, report the results and next section is discussion part.

Response 4: Thank you for the suggestions! We separated the discussion from the results.

Point 5:   Discussion - you should be stated what the findings of your research and find out similar or not similar (contrast) previous studies findings, as much as you have.

Response 5:Thank you for the insight and helpful suggestions! The previous literatures studied the effect of mandatory policy in general. We found few literatures on the role of supervision in policy implementation. We have tried our best to supplement the literatures according to the suggestions and emphasized it as much as possible.

Point 6:  Conclusion - Can be improved on the basis of research objectives. Also, you required to add section limitation & future research and policy implication e.g., theoretical and practical implications. 

Response 6: We appreciate your insight and very helpful suggestions! We added the policy recommendations and revised the conclusion according to the suggestions.

Special thanks to you for your good comments. 

Reviewer 4 Report

  • I suggest that instead of having the questions asked within text, the authors have the list of questions asked as part of the appendix.
  • How was sample size determined, from how many people and how representative is the sample? Does the sample represent at least 10% of sampled population? This to me is not clear and need to be clarified. 
  • When you talk about randomly asking people to complete a questionnaire in a shopping mall, what sampling method is this because to me this is not simple random sampling? How do you define the sampling method you have used in this study? With sample random sampling, you need a sampling frame and a table of random numbers which surely was not used in this study.
  • A pilot of 20 participants was conducted (for study 1 and 3). Where was this conducted? Did you conduct this in the same area where data was collected? Did you change some of the questions that were not clear? In other words, how has the pilot study helped you to improve the questionnaire items and structure? 
  • For study 3, it appears that the authors have used junior high school students as their participants. This raises intriguing questions: how did you deal with the ethical issues of using minors as your participants. Did you get permission from the school, Department, or from their parents? Who signed the consent form for them to participate in this study since they are below the age of 18? What were the reasons of using junior high school students in this study?
  • I am also failing to see the biographical data of the respondents, e.g. education level, age, etc. Do you think that this factors had influence on the results? The only thing that is appearing on tis study is gender. But again, you don't indicate if gender has influence on the results obtained. Similarly, it is not clear if age and education level of participants have influence on the results you obtained in this study. It will be important to include these aspects in your revision. 
  • The section on the discussion is weak and need to be improved. The author need to give case study material that support or refute their findings. At the moment this is totally lacking. It will also be interesting to indicate the countries where there is a mandatory policy with actual supervision that has led to long-term effectiveness of waste separation by people.  I expect to see more references used to validate or refute the findings of this study. 
  • The conclusions is not thoroughly supported by the results presented in the article. Thus, the conclusion need to be improved. In addition, recommendations need to be made for policy makers.  

Author Response

Response to Reviewer 4 Comments

Point 1:  I suggest that instead of having the questions asked within text, the authors have the list of questions asked as part of the appendix.

Response 1: Thank you for the suggestion! We have put the questions of Study 3 into the appendix.

Point 2: How was sample size determined, from how many people and how representative is the sample? Does the sample represent at least 10% of sampled population? This to me is not clear and need to be clarified. 

Response 2: Thank you for the suggestion! The sample size was estimated by G*Power (Version 3.1.9.2) (line 648-650). We added the demographic data from National Bureau of Statistics of China for comparison. The sample profile in this study is generally similar to the population profile of census.

Point 3: When you talk about randomly asking people to complete a questionnaire in a shopping mall, what sampling method is this because to me this is not simple random sampling? How do you define the sampling method you have used in this study? With sample random sampling, you need a sampling frame and a table of random numbers which surely was not used in this study.

Response 3: We are appreciative for your insightful suggestion! The sampling method was referenced from Wan’s study.

Wan, C., Shen, G.Q., Yu, A. (2014). The role of perceived effectiveness of policy measures in predicting recycling behaviour in Hong Kong. Resources, Conservation and Recycling, 83, 141-151. http://dx.doi.org/10.1016/j.resconrec.2013.12.009

Point 4:  A pilot of 20 participants was conducted (for study 1 and 3). Where was this conducted? Did you conduct this in the same area where data was collected? Did you change some of the questions that were not clear? In other words, how has the pilot study helped you to improve the questionnaire items and structure? 

Response 4: Thank you for pointing out the insufficiency of our research. Now we have added this information in the revised version. A pilot of 20 participants was conducted in Beijing (Study 1) and Dongguan (Study 3). Some wording in the questionnaire was then refined according to the results to improve the questionnaire’s items and structure.

Point 5: For study 3, it appears that the authors have used junior high school students as their participants. This raises intriguing questions: how did you deal with the ethical issues of using minors as your participants. Did you get permission from the school, Department, or from their parents? Who signed the consent form for them to participate in this study since they are below the age of 18? What were the reasons of using junior high school students in this study?

Response 5: We appreciate your detailed and very helpful suggestions! Before the study, we had obtained the consent of the school teachers and students complying with the ethical code of the Institute. The reasons of using junior high school students in this study have been interpreted in 2.3. The present studies (line 312-324).

Point 6: I am also failing to see the biographical data of the respondents, e.g. education level, age, etc. Do you think that these factors had influence on the results? The only thing that is appearing on this study is gender. But again, you don't indicate if gender has influence on the results obtained. Similarly, it is not clear if age and education level of participants have influence on the results you obtained in this study. It will be important to include these aspects in your revision. 

Response 6: We appreciate your detailed and very insight suggestions! The biographical data was supplemented in Table 3 and Table 5. The biographical variables, as covariates, had no significant effect on willingness to engage in waste separation behavior except age. The biographical variables were not our research variable, so we didn’t conduct the further research.

Point 7:  The section on the discussion is weak and need to be improved. The author needs to give case study material that support or refute their findings. At the moment this is totally lacking. It will also be interesting to indicate the countries where there is a mandatory policy with actual supervision that has led to long-term effectiveness of waste separation by people. I expect to see more references used to validate or refute the findings of this study.

Response 7: Thank you for the insight and helpful suggestions! The previous literatures studied the effect of mandatory policy in general. We found few literatures on the role of supervision in policy implementation. We have tried our best to supplement the literatures according to the suggestions and emphasized it as much as possible.

Point 8: The conclusions are not thoroughly supported by the results presented in the article. Thus, the conclusion needs to be improved. In addition, recommendations need to be made for policy makers. 

Response 8: Thank you for the suggestion! We have added the policy recommendations (line 794-805) and revised the conclusions according to the suggestion.

Special thanks to you for your good comments. 

Round 2

Reviewer 1 Report

The authors have successfully addressed and responded to all my comments, and I'm satisfied with the most of the changes. 

However, I still feel confused about the differences between non-supervised and volunteer policy. As mentioned in your policy recommendation paragraph, 

" the supervision such as volunteer guidance is important to ensure the implements of various policies. " and " The volunteer can guide residents to separate correctly and return the mixed waste. " 

Do you mean supervised by volunteers based on the regulation and guidelines? I'm not sure whether this is suitable for China, but I think paid job is more effective, or it can be responsible by the Food and Hygiene Department of China? 

Just would like to strengthen the applicability of the research outcomes. 

Author Response

Response: Thank you for the comment and clarification question!

According to your suggestions, we have revised the manuscript.

The “volunteer guidance” should be replaced by “specially-assigned person for guidance”, which is the paid job set by community committees. They will supervise residents’ waste separation behaviors based on the regulation and guidelines. The voluntary policy is defined as the policy without supervision. There is only regulation and guidelines but nobody supervise.

Please see the section of 6.4 policy recommendation:

“Second, the supervision such as specially-assigned person for guidance or monitoring is important to ensure the implements of various policies. The paid job is set to supervised residents’ waste separation behaviors based on the regulation and guidelines. The specially-assigned person can guide residents to separate correctly and return the mixed waste. ”

Reviewer 3 Report

Dear authors.,
Thank you for revising the manuscript by following comments and suggestions. However, I request you restructure how you present the methodology part, results, and discussion because you wrote the whole section method, design, etc. I suggest you read the standard format and manuscript  (refer https://www.mdpi.com/1660-4601/18/14/7308/htm) for your reference on how the paper structure is reported. Here is attached my comments and suggestions in pdf file. Please revise your manuscript carefully. Wish you luck. Thank you.

Author Response

Response: Thank you for the suggestion! We have revised the structure of the methodology part, results, and discussion in each study.

Thank you for give us the standard format manuscript for our reference.

Ibrahim, R. Z. A. R., Zalam, W. Z. M., Foster, B., Afrizal, T., Johansyah, M. D., Saputra, J., . . . Ali, S. N. M. (2021). Psychosocial work environment and teachers' psychological well-being: The moderating role of job control and social support. International Journal of Environmental Research and Public Health, 18, 7308. https://doi.org/10.3390/ijerph18147308

Reviewer 4 Report

The document has improved tremendously. I accept this manuscript subject to the following minor changes that need to be made before the document is published. 

  • I have come across a number of grammatical errors that need to improved. 
  • Since this research has been approved by the University of Chinese Academy of Sciences, Are you able to provide the ethics clearance number.
  • I am still failing to see case study materials on the general discussions of the findings. The author should have given case study material that support or refute their findings. At the moment this is still lacking. I was also expecting the authors to include the countries where there is a mandatory policy with actual supervision that has led to long-term effectiveness of waste separation by people. Unfortunately, this is still lacking.
  • Please, address my suggestion above to the satisfaction of the editor.

Author Response

Response: Thank you for the insight and helpful suggestions! The previous literature on the effect of mandatory policy is relatively general. We supplemented the literature about mandatory policy and supervision respectively and emphasized it as much as possible.

We refuted one study by Hao et al., (2020). Please see “6.1. Mandatory waste separation policies”: “A policy without supervision is as ineffective at encouraging waste separation as no policies. The finding is consistent with the study conducted by Hao et al. [24, 73], which found the waste separation rate was still low even under a policy due to a lack of clear supervision, intervention or punishment. ”

Additionally, we referred to the countries where there is a mandatory policy with actual supervision that has led to long-term effectiveness of waste separation by people. Please see “6.4. Policy recommendations”: Such as in the countries of Japan and Sweden, there are mandatory policies with actual supervision that have led to long-term effectiveness of waste separation by people.

Special thanks to you for your good comments.